

# AL360181.1 promotes proliferation and invasion in colon cancer and is one of ten m6A-related lncRNAs that predict overall survival

Yi Luo[1,*], Yayun Xie[1,*], Dejun Wu[2], Bingyi Wang[1], Helei Lu[1], Zhiqiang Wang[1], Yingjun Quan[1] and Bo Han[1]

[1] Tongren Hospital, Shanghai Jiao Tong University School of Medicine, Shanghai, China
[2] Shanghai Pudong Hospital, Fudan University Pudong Medical Center, Shanghai, China
[*] These authors contributed equally to this work.

## ABSTRACT

**Background**. N6-methyladenosine (m6A) exerted a pivotal role in colon cancer. Nevertheless, the long non-coding RNAs (lncRNAs) associated with this process have yet to be elucidated.

**Methods**. The open-access data used for analysis was downloaded from The Cancer Genome Atlas (TCGA) database for analysis, employing the R software for computational evaluations. The RNA level of specific molecules was assessed using the quantitative real-time PCR. CCK8, colony formation and transwell assay were used to evaluate the proliferation, invasion and migration ability of colon cancer cells.

**Results**. Here, we identified the m6A regulators from TCGA data and subsequently pinpointed lncRNAs with a |Cor| > 0.3 and $P < 0.05$, categorizing them as m6A-associated lncRNAs. Moreover, we formulated a prognosis signature rooted in ten m6A-related lncRNAs, consisting of AL360181.1, PCAT6, SNHG26, AC016876.1, AC104667.2, AL114730.3, LINC02257, AC147067.1, AP006621.3 and AC009237.14. This signature exhibited notable predictive accuracy in gauging patient survival. Immune-related evaluations revealed varied immune cell infiltration patterns across different risk groups, with our findings suggesting superior immunotherapy response in low-risk patients. Biological enrichment analysis indicated that the high-risk patients had a higher activity of multiple carcinogenic pathways, including glycolysis. The previously unreported lncRNA, AL360181.1, displayed a connection to glycolytic activity and diminished survival rates, warranting further investigation. The result indicated that AL360181.1 was correlated with more aggressive clinical characteristics. Immune infiltration assessments found AL360181.1 to have a positive correlation with Tcm infiltration, but an inverse relationship with entities like Th2 cells, T cells, neutrophils and macrophages. Biological enrichment analysis indicated that the pathways of WNT/β-catenin, pancreas beta cells, hedgehog signaling and some metabolism pathways were upregulated in high AL360181.1 patients. *In vitro* experiments showed that AL360181.1 was upregulated in the colon cancer cells. Moreover, AL360181.1 significantly promotes the proliferation, invasion and migration of colon cancer cells.

**Conclusions**. Our results can provide direction for future studies on m6A-related lncRNA in colon cancer.

Corresponding authors
Yingjun Quan,
qyj4347@shtrhospital.com
Bo Han, HB3616@shtrhospital.com

## INTRODUCTION

Colon cancer, a gastrointestinal malignancy, originates from the epithelium of the colon mucosa (*Benson et al., 2018*). Globally, it stands as a dominant malignancy, accounting for over a million new diagnoses annually (*Benson et al., 2018*; *Auclin et al., 2017*). As a multifactorial disease, the incidence rate of colon cancer has great regional differences, often linked to lifestyle factors. Additionally, the incidence rate of men is higher than that of women, and it increases significantly with age (*Weitz et al., 2005*). Surgery combined with neoadjuvant/adjuvant therapy remains the cornerstone of colon cancer treatment (*Biondo et al., 2019*). For those diagnosed in the non-advanced stage, the five-year survival rate post-standard surgical intervention can exceed 90% (*Khan & Cahill, 2021*). However, approximately 20% of patients, due to a dearth of early symptoms, are diagnosed at advanced stages, compromising the ideal surgical window owing to local progression or distant metastases (*Biller & Schrag, 2021*). Consequently, it is significant to strengthen the research on the pathogenesis and development of colon cancer in terms of tumor molecular mechanism and to explore more effective therapeutic targets.

Tumors frequently exhibit epigenetic anomalies, recognized as a hallmark of cancer (*Hanahan, 2022*). N6-methyladenosine (m6A) epitomizes a prevalent epigenetic alteration (*He et al., 2019*). Typically, m6A dynamics are orchestrated by three kinds of molecules, m6A writers, erasers and readers (*Wang et al., 2020c*). *Han et al. (2019)* observed that pri-miR221/222 maturation could be accelerated by METTL3, therefore promoting bladder cancer progression. *Huang et al. (2022)* illustrated that the APOE mRNA stability could be decreased by FTO in an N6-methyladenosine-dependent manner, thereby repressing glycolysis and tumor growth. *Zhang et al. (2021)* noticed that IGF2BP1 could boostglucose metabolism and colon cancer progression by enhancing mRNA stability. Within the realm of RNA, long non-coding RNAs (lncRNAs) are distinguished by their length, surpassing 200 nucleotides (*Bhan, Soleimani & Mandal, 2017*; *Yokoi & Nakagawa, 2022*). There are various biological processes that lncRNA participates in and regulates, also including the m6A process. For exmaple, *Liu et al. (2022)* highlighted how METTL3-mediated m6A modifications bolster lncRNA THAP7-AS1, pivotal for CUL4B's nuclear translocation and gastric cancer evolution. In colon cancer, *Hou et al. (2021)* found that the complex of LINC00460/ DHX9/IGF2BP2 could facilitate cancer proliferation and invasion by regulating HMGA1 mRNA stability depending on the m6A process.

The progress of bioinformatics provides reliable tools for researchers to recognize diseases (*Zhang et al., 2022*; *Jiang et al., 2021*; *Liu et al., 2021*). In our study, we delved deeply into the lncRNAs associated with colon cancer. Moreover, we formulated a prognosis signature using ten m6A-related lncRNAs, consisting of AL360181.1, PCAT6, SNHG26, AC016876.1, AC104667.2, AL114730.3, LINC02257, AC147067.1, AP006621.3 and AC009237.14. Subsequent biological enrichment and immune-related analysis were conducted on patients from different risk groups. The lncRNA AL360181.1, previously

unreported, was identified to have a strong association with glycolytic activity and poorer survival rates, prompting further investigation. *In vitro* tests revealed an elevation in AL360181.1 levels within colon cancer cells, and crucially, this lncRNA markedly enhanced the proliferation, invasion, and migration of these cells.

## METHODS

Portions of this text were previously published as part of a preprint (*Luo et al., 2023*).

### Collection of open-accessed data

The transcriptional profile and clinical feature data were directly downloaded from The Cancer Genome Atlas (TCGA) database through the TCGA-GDC project. The R software facilitated the acquisition and assembly of this raw data. Probe annotation were conducted using the human GRCh38 file, and data preprocessing was accomplished using the limma, dplyr, and tidyr packages. The m6A related molecules were collected from previous studies, including ALKBH5, METTL3, FTO, METTL14, HNRNPC, WTAP, YTHDF2, RBM15, YTHDF1, ZC3H13, YTHDC2, YTHDC1.

### Identification of m6A-related lncRNAs

LncRNAs with a median value of less than 0.5 across all samples were excluded. Subsequent, correlation analysis pinpointed the lncRNAs with $|Cor| > 0.3$ and $P < 0.05$ of m6A molecules, designating them as m6A-related lncRNAs. The generated mRNA-lncRNA network was accomplished in the Cytoscape software.

### Prognosis evaluation

Univariate Cox regression analysis was conducted on input genes to identify the molecules significantly correlated with patients survival with $P < 0.1$. Subsequently, LASSO regression analysis was performed for data dimension reduction. A prognosis model was crafted through multivariate Cox regression analysis for prognosis construction with the formula of "Risk score = lncRNA A * Coeff A + lncRNA B * Coeff B + … + lncRNA N * Coeff N". The evaluation of the prognosis signature was completed using the KM and ROC curves.

### Immune microenvironment analysis

Quantitative assessment of the colon cancer immune microenvironment leveraged algorithms like XCELL, TIMER, QUANTISEQ, EPIC, MCPCOUNTER and CIBERSORT algorithms (*Aran, Hu & Butte, 2017*; *Plattner, Finotello & Rieder, 2020*; *Li et al., 2017*; *Racle & Gfeller, 2020*; *Becht et al., 2016*; *Chen et al., 2018*). Immunotherapy response was evaluated using the Tumor Immune Dysfunction and Exclusion (TIDE) algorithm (*Fu et al., 2020*).

### Biological enrichment

To uncover potential biological disparities, we employed the Gene Set Enrichment Analysis (GSEA) based on a designated pathway reference file (*Subramanian et al., 2005*). The ClueGO app aided in the enrichment and depiction of biological functions within the Cytoscape software framework. Additionally, the Gene Set Variation Analysis (GSVA) was harnessed for pathway quantification.

## Drug sensitivity analysis

Analysis of drug sensitivity was based on the data from the Genomics of Drug Sensitivity in Cancer (GDSC) database (*Yang et al., 2013*).

## Cell lines

The human normal colon epithelial cells (NCM460) and three colon cancer cells (HCT116, SW480 and DLD-1) were purchased from the cell bank of the Chinese Academy of Sciences in Shanghai. Cells were maintained in standard culture conditions (37 °C, 5% CO2), with passages carried out every 3–4 days.

## Real-time Quantitative PCR (qRT-PCR)

We extracted total RNA using a total RNA extraction kit, which was then reverse-transcribed into cDNA for subsequent assays. The process of qRT-PCR was performed according to established protocols (*Wu et al., 2023*). The primer used was as follows: AL360181.1, forward, 5′-ACACCACTGTCAGCACAGA-3′, reverse, 5′-ATTATCTAAAGACCTGGGCCT-3′; GAPDH, forward, 5′-GGAGCGAGATCCCTCCAAAAT-3′, reverse, 5′-GGCTGTTGTCAT ACTTCTCATGG-3′.

## Cell transfection

Cell transfection was performed using the lipofectamine 2,000 according to the standard procedure. The sequence used for cell transfection were as follows: sh#1: CCACTAGCACATGGAGTCA; sh#2: AGTCACCTCTGGCCTTCAA; sh#3: AGGCCCAGGTCTTTAGATA.

## *In vitro* experiments

Cell proliferation capacity was assessed using the CCK8 and colony formation assays. Cell invasion and migration ability were evaluated using the transwell assay. All the assays were performed based on the standard procedure (*Wu et al., 2023*).

## Statistical analysis

Public data-based analyses were executed with the R software, using a significance threshold set at 0.05. Depending on the data distribution, appropriate statistical methods were applied.

# RESULTS

The complete workflow is depicted in Fig. S1.

## Identification of m6A-related lncRNAs

Initially, we analyzed the expression levels of m6A regulators in colon cancer, including ALKBH5, METTL3, FTO, METTL14, HNRNPC, WTAP, YTHDF2, RBM15, YTHDF1, ZC3H13, YTHDC2, YTHDC1. Our findings demonstrated that the majority of these regulators were markedly elevated in colon cancer tissues, underscoring their pivotal role in carcinogenesis (Figs. 1A–1D). Additionally, correlation analysis identified the lncRNAs with $|Cor| > 0.3$ and $P < 0.05$ of m6A molecules, defined as m6A-related lncRNAs (Fig. 1E).
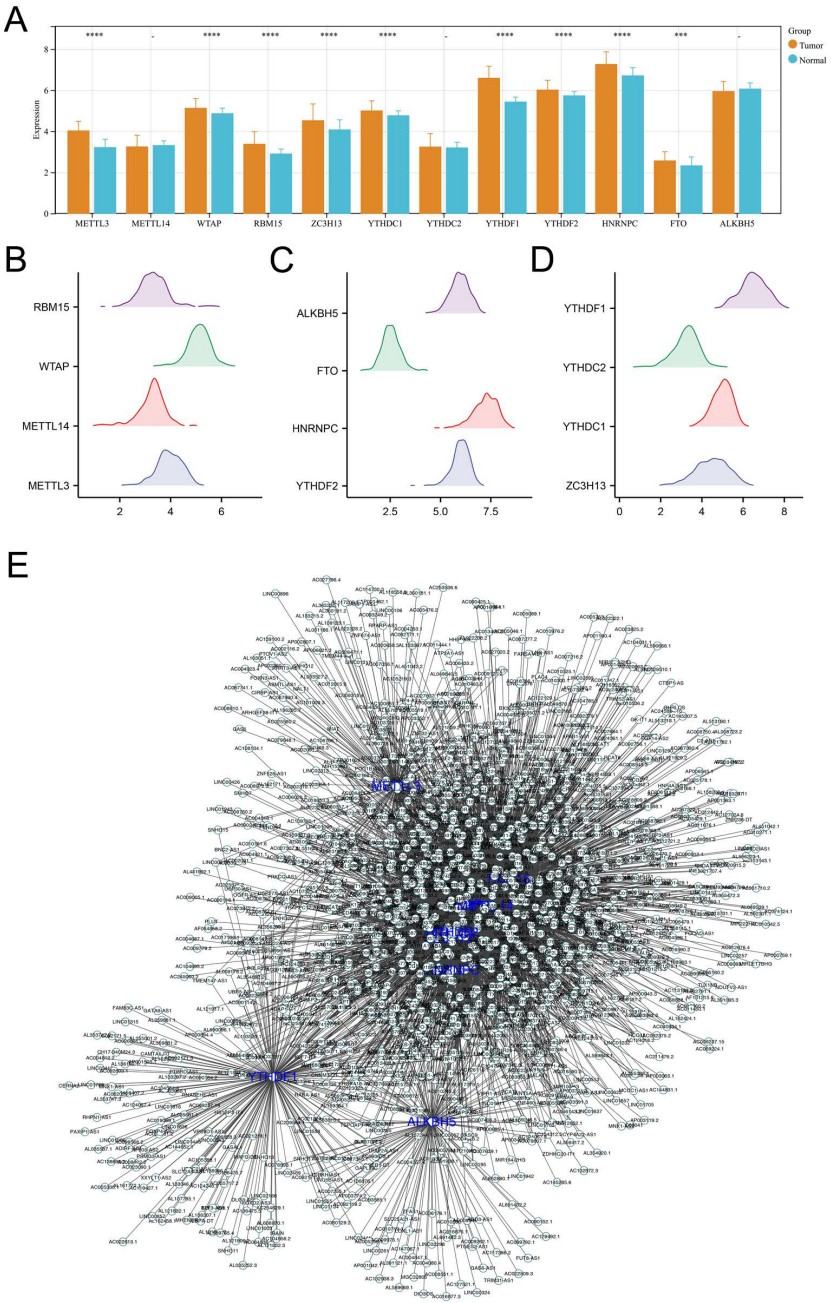

**Figure 1 Identification of m6A-related lncRNAs in colon cancer.** (A) The expression level of m6A regulators in colon cancer and adjacent tissue, ns = $P > 0.05$, *** = $P < 0.001$, **** = $P < 0.0001$; (B–D) The expression pattern of m6A regulators in colon cancer; (E) The lncRNAs with |Cor| > 0.3 and $P < 0.05$ of these m6A molecules were identified and defined as m6A-related lncRNAs.

## The prognosis signature was established based on m6A-related lncRNAs

Subsequently, the expression value of these m6A-related lncRNAs was extracted and integrated with patients survival information. Using a univariate Cox regression, we pinpointed lncRNAs with significant ties to patient survival, showcasing the top 50 in Fig. 2A. LASSO regression analysis was then employed to refine optimize variables (Figs. 2B–2C). Ultimately, the multivariate Cox regression analysis was conducted and ten lncRNAs were identified for prognosis signature, consisting of AL360181.1, PCAT6, SNHG26, AC016876.1, AC104667.2, AL114730.3, LINC02257, AC147067.1, AP006621.3 and AC009237.14 (Fig. 2D). Following, we evaluated the performance of the prognosis signature. Our assessment of this signature revealed that high-risk patients likely faced inferior survival outcomes in the training cohort. Additionally, the ROC curves affirmed the reliability of our prognostic model in predicting patient survival (Fig. 3A), a finding consistent in the validation cohort (Fig. 3B). Clinical correlation analysis indicated that PCAT6 was overexpressed in patients with <= 65 features (Fig. 3C); AC009237.14 was upregulated in female patients (Fig. 3D); AL360181.1, PCAT6, LINC02257, AP006621.3, AC009237.14 and risk score were upregulated in patients with stage III-IV (Fig. 3E); AL360181.1, LINC02257 and risk score were overexpressed in patients with T3-4 (Fig. 3F); AL360181.1, PCAT6, AC009237.14 and risk score were upregulated in patients with distant metastasis (Fig. 3G); PCAT6, AC104667.2, LINC02257, AC009237.14 and risk score were overexpressed in patients with N1-3 stage (Fig. 3H). Additionally, we designed a nomogram integrating the risk score with various clinical parameters (Fig. S2).

## Immune microenvironment analysis

We proceeded to assess the distinctions in the immune microenvironment between high- and low-risk patient groups To quantify the immune landscape in colon cancer, we employed algorithms such as XCELL, TIMER, QUANTISEQ, EPIC, CIBERSORT, and MCPCOUNTER, illustrated in Fig. 4A. A positive correlation was found between risk score and Tregs, endothelial cells, B cells, and macrophages of type M2, while resting myeloid dendritic cells, neutrophils, and CD4+ memory T cells were in contrast (Figs. 4B–4C). Moreover, high-risk patients exhibited elevated expression of critical immune checkpoints like PDCD1, PDCD1LG2, CTLA4, and CD274 (Fig. 4D). Notably, TIDE analysis revealed increased TIDE scores and CAF infiltration in these patients, suggesting a less favorable response to immunotherapy (Fig. 4E).

## Biological enrichment and drug sensitivity analysis

Then, we delved into the potential biological variances. ClueGO analysis highlighted enriched terms of regulation of neuron migration, phosphorylative mechanism, positive regulation of synapse assembly, positive regulation of blood vessel, myofibril assembly, negative regulation of blood circulation and neuron maturation were significantly enriched (Fig. 5A). GSVA analysis pinpointed the activation of glycolysis, KRAS signaling, hypoxia, TGF-beta signaling and epithelial-mesenchymal transition (EMT) were activated in the high-risk patients (Fig. 5B). Using the KEGG gene set, GSEA analysis underscored the

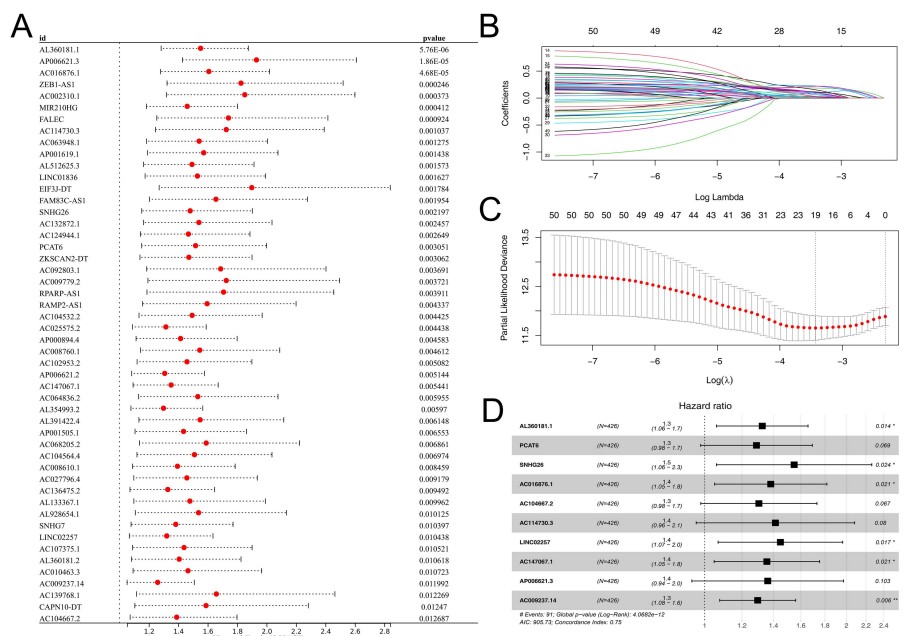

**Figure 2** **Prognosis signature.** (A) Univariate Cox regression analysis was conducted to identify the m6A-related lncRNAs associated with patients survival; (B–C) LASSO regression analysis; (D) Multivariate Cox regression analysis.

involvement of neuroactive ligand–receptor interaction, olfactory transduction, and calcium signaling pathways (Fig. 5C). Drug sensitivity analysis indicated that imatinib and pazopanib might be more effective in low-risk patients, while bosutinib might be more effective for high-risk patients (Fig. 6).

## Further exploration of AL360181.1

We further assessed the pathway activity of glycolysis in colon cancer (Fig. 7A). Data revealed that the model lncRNAs AL360181.1, SNHG26, LINC02257, AC147067.1 and risk score had a higher level in patients with high glycolysis activity (Fig. 7B). KM survival data underscored that higher AL360181.1 and LINC02257 levels correlated with poorer survival outcomes, while SNHG26 and AC147067.1 showed no significant difference (Figs. 7C–7F). The lncRNA AL360181.1 has not been reported previously. Meanwhile, the lncRNA AL360181.1 was selected for further analysis for its correlation with glycolysis activity and worse survival performance. Clinical correlation analysis indicated that the patients with worse clinical features (TNM classification and lymphatic invasion) might have a higher AL360181.1 expression (Figs. 7G–7J). The molecules correlated with AL360181.1 was shown in Fig. 7K. The result of ssGSEA revealed a positive relationship between AL360181.1 and Tcm infiltration but a negative one with entities like Th2 cells, neutrophils, iDC, macrophages, and T cells (Fig. 7L). Biological enrichment analysis indicated that WNT-β catenin, pancreas beta cells, hedgehog signaling, and some metabolism pathways were activated in the patients with high AL360181.1 expression (Fig. 8). Clinical data also associated higher AL360181.1 levels with advanced clinical stages. Therefore, we selected

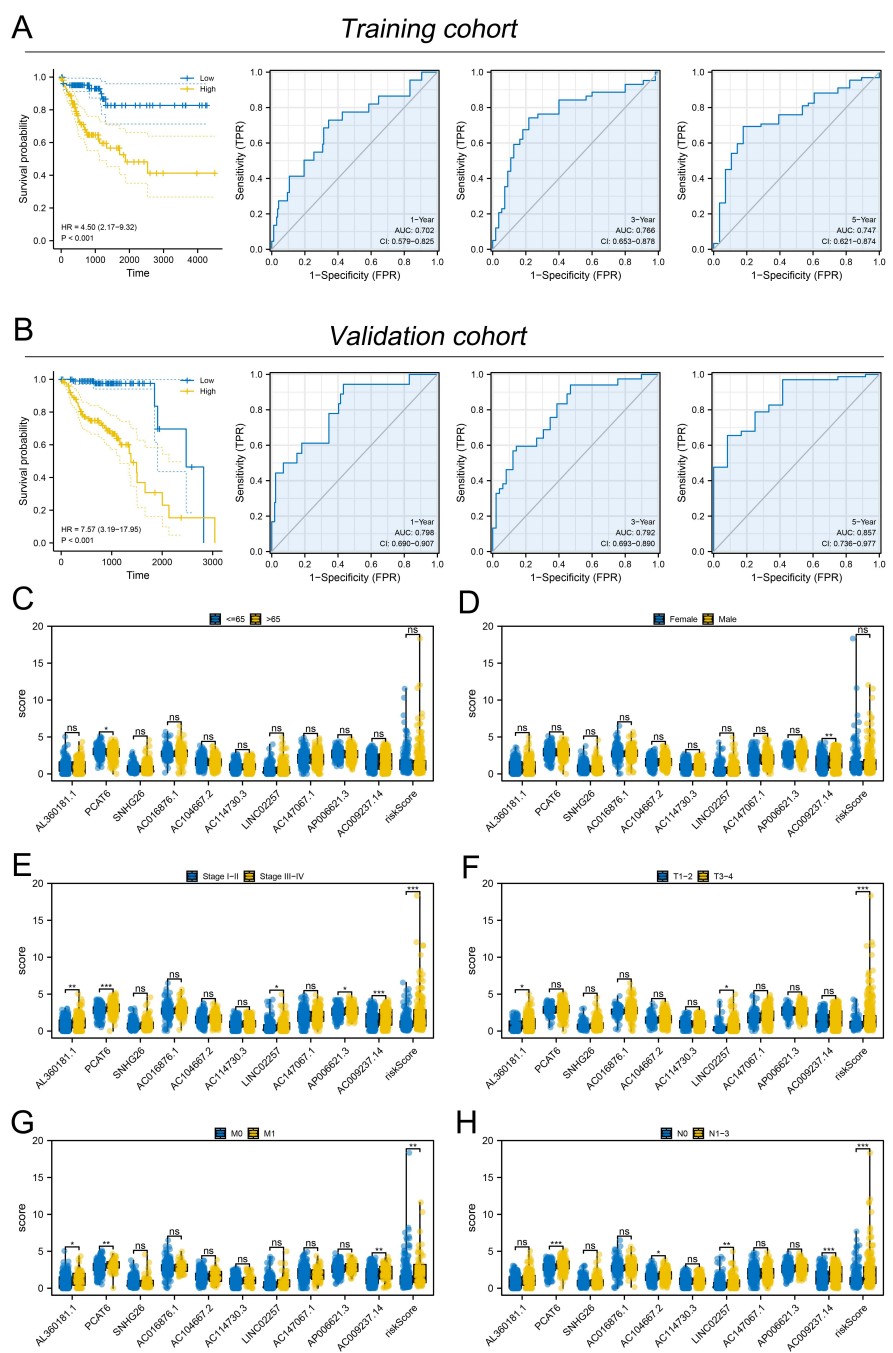

**Figure 3** **Evaluation of the performance of prognosis signature.** (A) KM and ROC curves of our model in the training cohort; (B) KM and ROC curves of our model in the validation cohort; (C–H) Clinical correlation of model lncRNAs and risk score, ns = $P > 0.05$, * = $P < 0.05$, ** = $P < 0.01$, *** = $P < 0.001$.

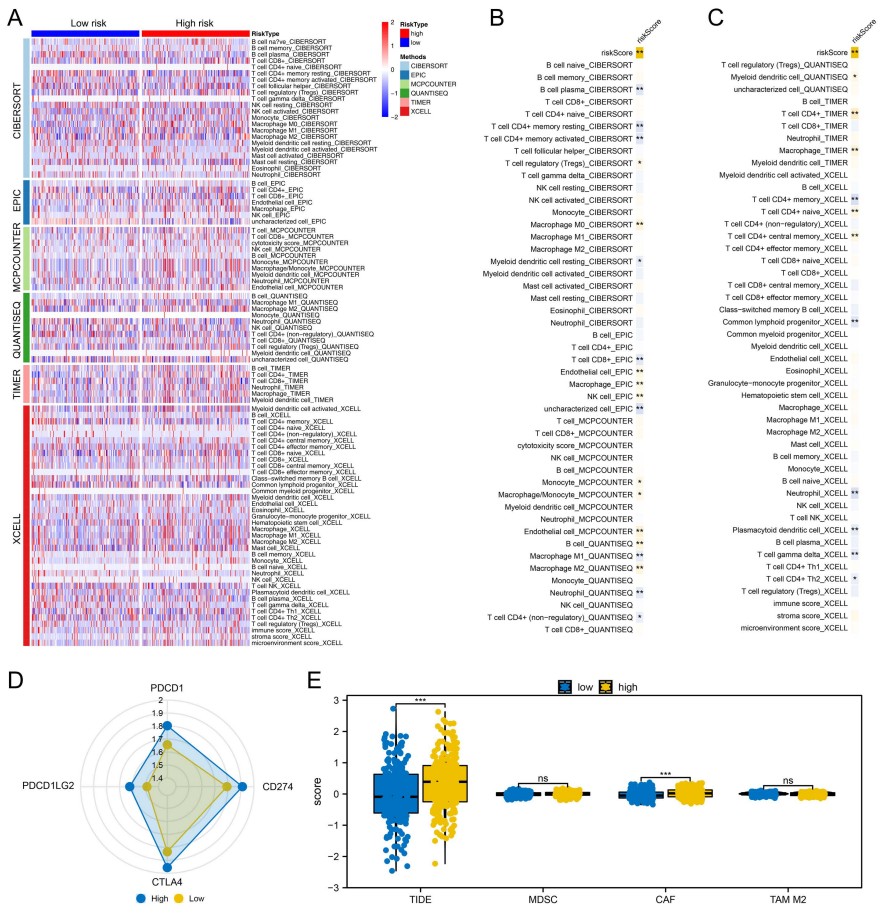

**Figure 4** **Immune-related analysis.** (A) Multiple algorithms were performed to quantify the immune microenvironment of colon cancer; (B–C) Correlation between the risk score and quantified immune cells, * = $P < 0.05$, ** = $P < 0.01$; (D) The key immune checkpoint level in high and low-risk patients; (E) The expression level of TIDE, MDSC, CAF and M2 TAM in high- and low-risk patients, ns = $P > 0.05$, *** = $P < 0.001$.

AL360181.1 for the subsequentexperiments. Results of qRT-PCR showed that AL360181.1 was upregulated in the colon cancer cells, especially in the DLD-1 and SW480 cells (Fig. 9A). Then, the AL360181.1 was knocked down in the DLD-1 and SW480 cells to explore its effect on biological behavior. After silencing AL360181.1 in these cells, sh#2, with the highest knockdown efficiency, was chosen for further studies (Figs. 9B–9C). CCK8 and colony formation assay showed that the knockdown of AL360181.1 significantly inhibits the cell proliferation ability of colon cancer cells (Figs. 9D–9E). Results of the transwell assay indicated that the inhibition of AL360181.1 could remarkably suppress the invasion and migration ability of colon cancer cells (Fig. 9F).

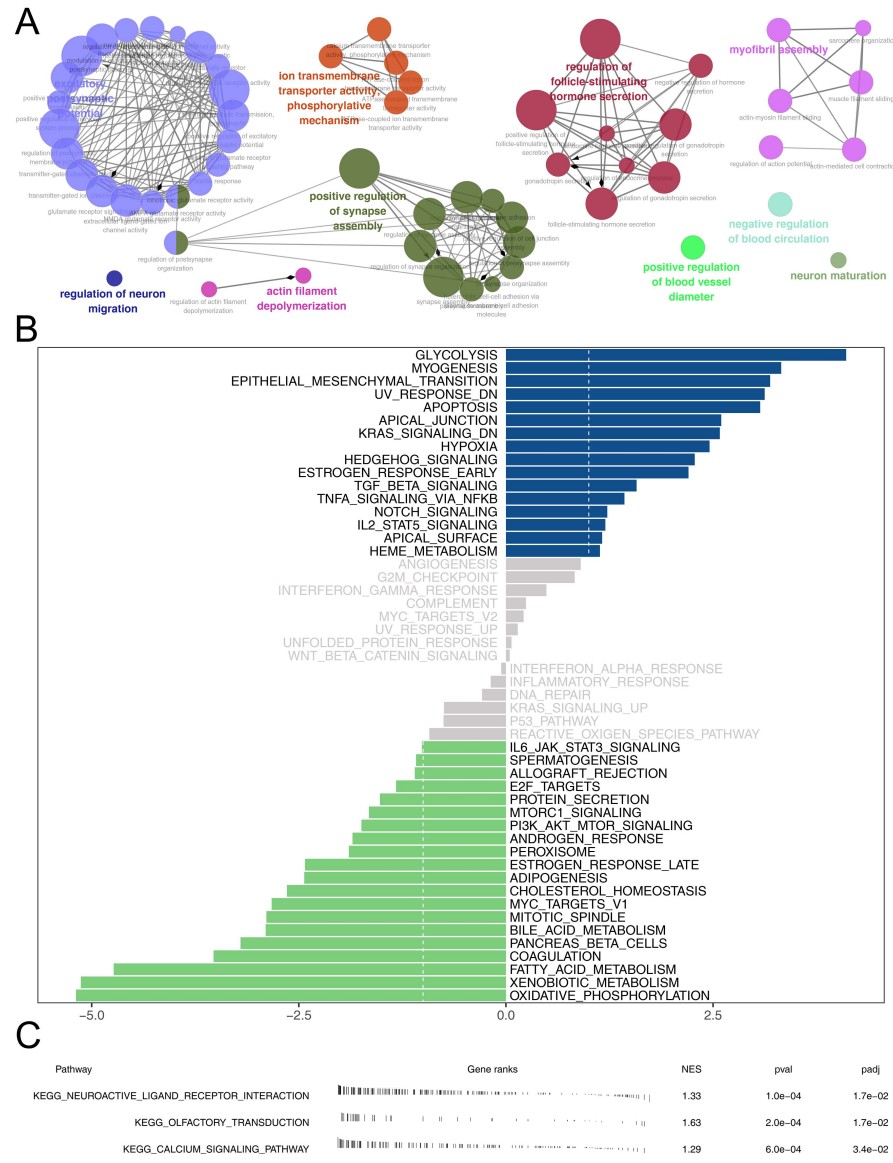

**Figure 5  Biological enrichment analysis.** (A) ClueGO analysis; (B) GSVA analysis between high and low-risk patients; (C) GSEA analysis based on KEGG gene set between high and low-risk patients.

# DISCUSSION

Colon cancer continues to be a pressing global health concern (*Labianca et al., 2010*). Beyond adding to the societal medical burden, it profoundly impacts the well-being, mental health, and life expectancy of patients (*Dienstmann, Salazar & Tabernero, 2015*). Therefore, it is very important to deeply explore the pathogenesis of colon cancer from a biological perspective to identify and develop new clinical targets.

Here, we extensively analyzed lncRNAs associated with colon cancer. Moreover, we established a prognosis signature based on ten m6A-related lncRNAs, consisting of

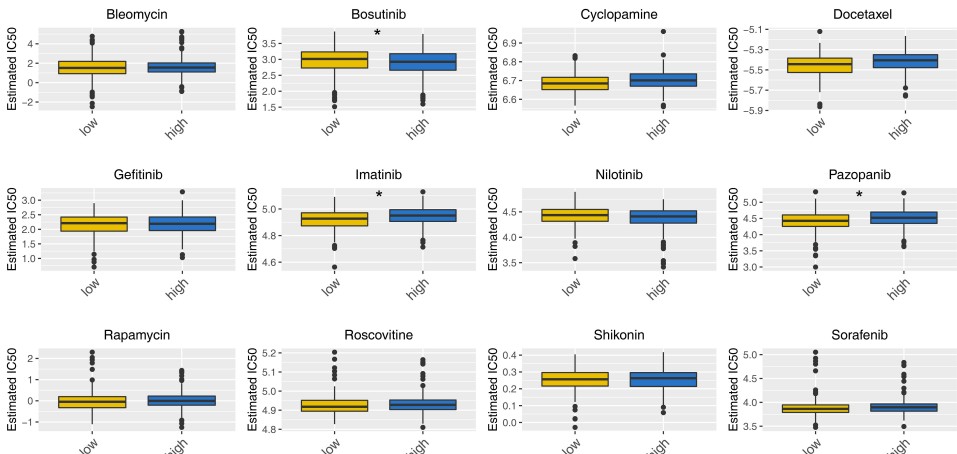

**Figure 6** **Drug sensitivity analysis between high- and low-risk patients.** *= $P < 0.05$.

AL360181.1, PCAT6, SNHG26, AC016876.1, AC104667.2, AL114730.3, LINC02257, AC147067.1, AP006621.3 and AC009237.14, which showed satisfactory prediction efficiency in patients survival. Immune-related analysis indicated a diverse immune cell infiltration pattern between high- and low-risk patients. Also, we found that low-risk patients might be more sensitive to immunotherapy. Biological enrichment analysis indicated that the high-risk patients had a higher activity of multiple carcinogenic pathways, including glycolysis. The lncRNA AL360181.1 has not been reported previously. Meanwhile, the lncRNA AL360181.1 was correlated with glycolysis activity and worse survival performance, therefore selected for further analysis. *In vitro* experiments showed that AL360181.1 was upregulated in the colon cancer cells. Moreover, AL360181.1 significantly promotes the proliferation, invasion and migration of colon cancer cells.

Our findings reinforce the vital association between model lncRNAs and patient outcomes. Some of these lncRNAs have been spotlighted in other cancers. *Dong et al. (2020)* highlighted lncRNA PCAT6's role in augmenting breast cancer progression through VEGFR2 modulation. *Lang et al. (2021)* found that the lncRNA PCAT6 could enhance IGF1R mRNA stability through the PCAT6/IGF2BP2/IGF1R RNA-protein complex, which was induced in an m6A manner. In tongue squamous cancer, *Jiang et al. (2022)* noticed that the lncRNA SNHG26 could enhance cell malignant biological behaviors and cisplatin resistance of cancer by affecting the PGK1/Akt/mTOR signaling. Our insights into m6A-related lncRNAs in colon cancer shed light on their mechanistic roles, paving the way for future research avenues.

Our result also suggested that the pathway of glycolysis, EMT, KRAS signaling, hypoxia, and TGF-beta signaling were heightened in high-risk patients. Previous studies have focused on the role of these pathways in cancer. In pancreatic cancer, *Hu et al. (2019)* discerned UHRF1's role in amplifying glycolysis and proliferation by inhibiting SIRT4 expression. The EMT pathway is an important process for tumor invasion and metastasis. *Wang et al. (2020a)* discovered that the cinobufacini could suppress the invasion of colon cancer cells

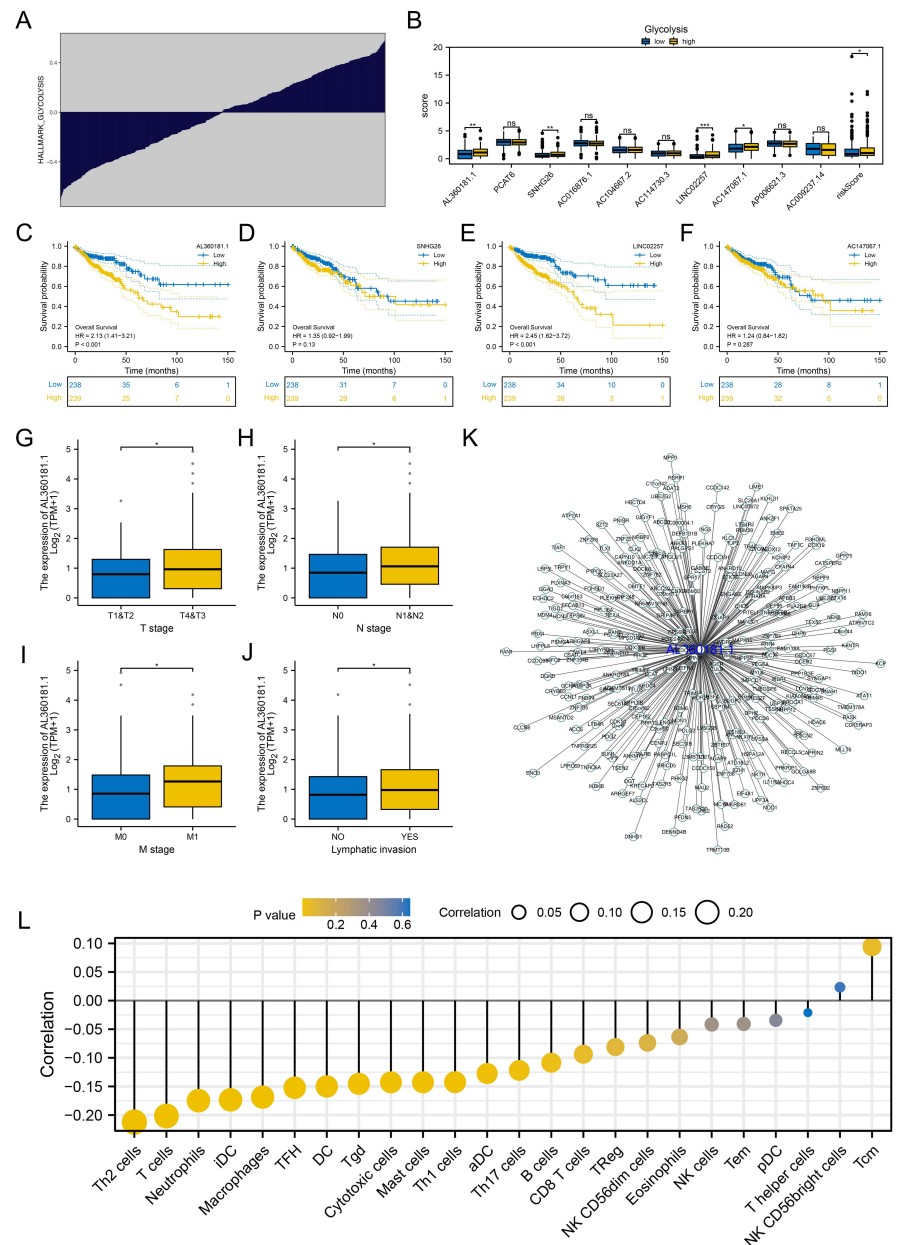

**Figure 7 Further exploration of AL360181.1.** (A) Quantification of the glycolysis activity in colon cancer; (B) The expression level of model lncRNAs and risk score in patients with high and low glycolysis activity, ns = $P > 0.05$, * = $P < 0.05$, ** = $P < 0.01$, *** = $P < 0.001$; C–F KM survival curves of AL360181.1, SNHG26, LINC02257, AC147067.1; (G–J) The expression level of AL360181.1 in patients with different clinical stage, *= $P < 0.05$; (K) The molecules regulated by AL360181.1; (L) The correlation between AL360181.1 and specific immune cells.

**Figure 8** Biological enrichment analysis of AL360181.1.

by targeting Wnt/β-Catenin and EMT signaling. Remarkably, we noticed that the lncRNA AL360181.1, SNHG26, LINC02257 and AC147067.1 are elevated in patients manifesting heightened glycolysis activity, suggesting their potential role in colon cancer progression *via* glycolytic modulation.

Diving into immune infiltration, our analysis uncovers a positive association between the risk score and Tregs, endothelial cells, B cells, and M2 macrophages, contrasted by its negative relationship with CD4+ memory T cells, neutrophils, and resting myeloid dendritic cells. Conventionally, Tregs inhibit killing immunity in the tumor microenvironment and are regarded as cancer-promoting cells. In colon cancer, *Sui et al. (2020)* noted the influence of YYFZBJS on gut microbiota alterations, which consequently reduces Tregs infiltration, accelerating colon cancer progression. Moreover, *Cheng et al. (2018)* observed that PKN2 could regulate the polarization of the M2 macrophages through the DUSP6-Erk1/2 pathway, further affecting tumor progression. Glycolysis is a fundamental cellular metabolic process, but in cancer it plays a special role (*Ganapathy-Kanniappan & Geschwind, 2013*). Cancer cells often exhibit high rates of glycolysis even in the presence of oxygen, a phenomenon known as the "Warburg effect" (*Liberti & Locasale, 2016*). Glycolysis not only provides cancer cells with rapid energy and metabolic intermediates required for growth, but it also plays an important role in the tumor immune microenvironment (*Gatenby & Gillies, 2004*). First, lactic acid produced by glycolysis can lead to acidification of the tumor microenvironment, which in turn helps cancer cells avoid being cleared by the immune system (*Li et al., 2020*; *Wang et al., 2018*). In addition, lactic acid can also inhibit the activity of certain immune cells, such as T cells and natural killer cells, further helping cancer cells to evade the attack of the immune system (*Wang et al., 2020b*).

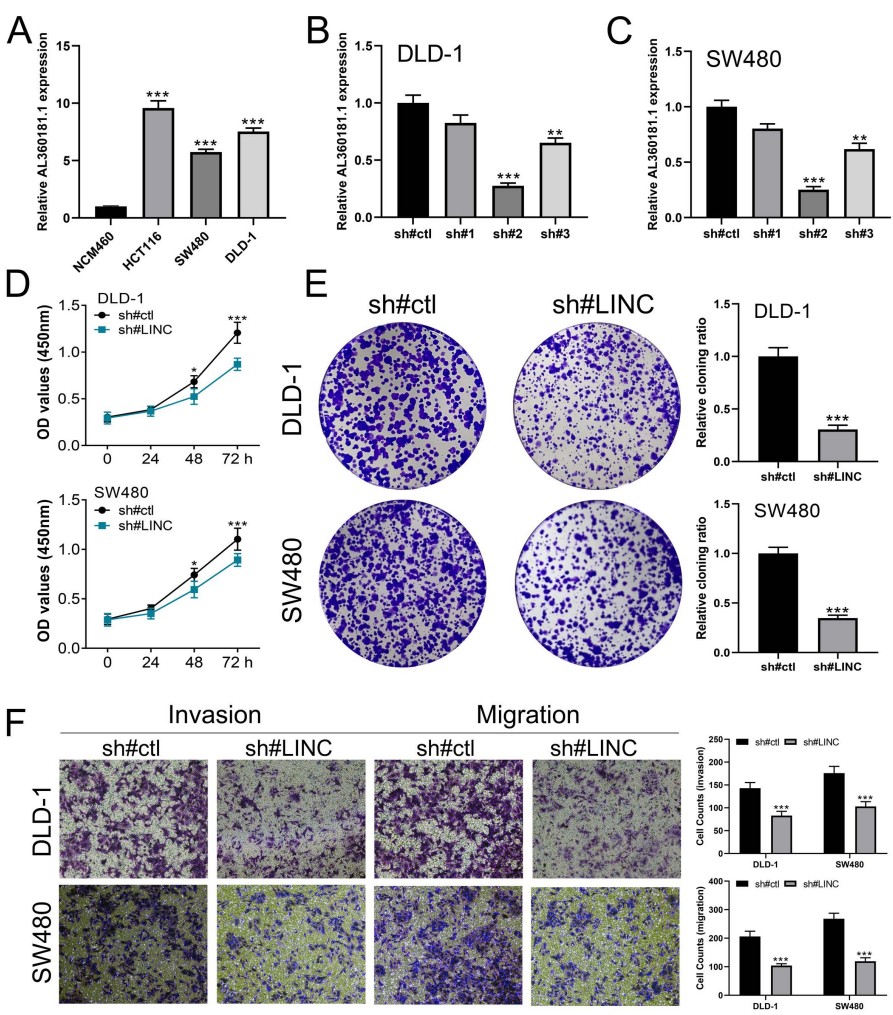

**Figure 9** *In vitro* experiments of AL360181.1 in colon cancer cells. (A) Expression level of AL360181.1 in colon cancer cells, *** = P < 0.001; (B–C) qRT-PCR was used to evaluate the knockdown efficiency of AL360181.1 in colon cancer, ** = P < 0.01, *** = P < 0.001; (D, C) CK8 assay was performed in the sh# ctl and sh# AL360181.1 cells, * = P < 0.05, *** = P < 0.001; (E) Colony formation assay was performed in the sh#ctl and sh#AL360181.1 cells, *** = P < 0.001; (F) Transwell assay was performed in the sh#ctl and sh# AL360181.1 cells, *** = P < 0.001.

While our study is grounded in robust data and analysis, certain constraints cannot be overlooked. The primary datasets predominantly represent Western populations, potentially introducing bias and limiting general applicability, given racial biological variances. Also, our conclusions stem largely from bioinformatics, necessitating further comprehensive biological validations.

### Funding

This study was supported by the Initial Scientific Research Fund of Shanghai Tongren Hospital (TR2020rc01); the Research Fund of Key laboratory for translational research and innovative therapeutics of gastrointestinal oncology (NO. ZDSYS-2020-01); and the Scientific research project of Changning District Health Committee (NO. 20214Y011). The funders had no role in study design, data collection and analysis, decision to publish, or preparation of the manuscript.

### Grant Disclosures

The following grant information was disclosed by the authors:
Initial Scientific Research Fund of Shanghai Tongren Hospital: TR2020rc01.
Research Fund of Key laboratory for translational research and innovative therapeutics of gastrointestinal oncology:  ZDSYS-2020-01.
Scientific research project of Changning District Health Committee: 20214Y011.

### Competing Interests

The authors declare there are no competing interests.

### Author Contributions

- Yi Luo performed the experiments, analyzed the data, prepared figures and/or tables, authored or reviewed drafts of the article, and approved the final draft.
- Yayun Xie performed the experiments, analyzed the data, prepared figures and/or tables, and approved the final draft.
- Dejun Wu performed the experiments, analyzed the data, prepared figures and/or tables, and approved the final draft.
- Bingyi Wang performed the experiments, prepared figures and/or tables, authored or reviewed drafts of the article, and approved the final draft.
- Helei Lu performed the experiments, analyzed the data, authored or reviewed drafts of the article, and approved the final draft.
- Zhiqiang Wang analyzed the data, authored or reviewed drafts of the article, and approved the final draft.
- Yingjun Quan conceived and designed the experiments, authored or reviewed drafts of the article, and approved the final draft.
- Bo Han conceived and designed the experiments, authored or reviewed drafts of the article, and approved the final draft.

### Data Availability

The raw data is available at figshare: Wu, Dejun (2023). PeerJ raw data. figshare. Dataset. https://doi.org/10.6084/m9.figshare.23926089.v1.

## Supplemental Information

Supplemental information for this article can be found online at http://dx.doi.org/10.7717/peerj.16123#supplemental-information.

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
