# Peer review of "AL360181.1 promotes proliferation and invasion in colon cancer and is one of ten m6A-related lncRNAs that predict overall survival"

_PeerJ, doi:10.7717/peerj.16123_

## Round 0.1 · original submission · Major Revisions

This study explored the role of m6A-related lncRNAs, especially AL360181.1 in colon cancer with novelty. However, this paper's English is a little bit poor. The authors should explain why they used LASSO to make the model. What about other machine-learning techniques? Please make revisions according to the comments.

Reviewer 1 ·

Basic reporting

The paper named “AL360181.1 promotes proliferation and invasion in colon cancer and is one of ten m6A-related lncRNAs that predict overall survival” conducted by Luo et al. has been evaluated. I think this study has certain innovation and clinical significance. However, some obvious limitations impair the overall quality. Before it is suitable for publication, I suggest the author make substantive modifications.
1. Detailed description of all method section is required.
2. State the * and statistic value in the method section or otherwise mention in every figure legend. e.g., *p>0.05.
3. Check the reference style and abbreviation of journals.
4. Some typos are there; rectify those (Minor).
5. The relationship between immune cell infiltration and specific pathways should be well discussed in the relevant section.
6. The author constructed a prognostic signature. I suggest the author further construct a nomogram diagram to enhance its clinical application capabilities.

Experimental design

This study was conducted based on open-accessed data. Some previous studies have used similar bioinformatics or statistical methods. Although it will lose some innovation, I think it will benefit the reader if the authors could include the following references, which would provide readers with a more complete understanding of the whole study.

Validity of the findings

I noticed that some of the figures have a low resolution which makes them hard to read. I would suggest that the authors increase the resolution of these figures, which would make them easier to read and understand. This would also make it easier for readers to fully understand the results and make use of it. I believe that increasing the resolution of these figures would greatly improve the manuscript.

Reviewer 2 ·

Basic reporting

It is very honored to be invited the reviewing the manuscript entitled “AL360181.1 promotes proliferation and invasion in colon cancer and is one of ten m6A-related lncRNAs that predict overall survival”. As a whole, the author comprehensively explored the role of m6A-related lncRNAs, especially AL360181.1 in colon cancer. The manuscript is novel. Besides, they used many interesting methods. I think the manuscript is within the scope. Some problems should be solved.
Firstly, what are the criteria for the author to define m6A related lncRNAs? Is this standard reported?
Secondly, in the whole manuscript, there are many errors, which should be checked. The English writing level should be improved.
Thirdly, the author used LASSO regression to construct a prognostic model. What are the benefits of doing so? The author should elaborate specifically on it.
Fourthly, although all the methods is reasonable, we have to notice that the analysis has its limitations. The authors should provide the limitations in the discussion part with the following references:
Fifthly, what is the GDSC database? What specific data does it provide?
Sixthly, I think the description in the Figure legend section is still not detailed enough.

Experimental design

Within the scape

Validity of the findings

They validated the role of AL360181.1

Additional comments

None.

Reviewer 3 ·

Basic reporting

I have carefully reviewed the manuscript entitled " AL360181.1 promotes proliferation and invasion in colon cancer and is one of ten m6A-related lncRNAs that predict overall survival", and I appreciate the work and effort put into the research. However, I do believe the manuscript requires some modifications before it can be considered for publication in. My specific comments and suggestions are as follows:
1. There are numerous instances throughout the manuscript where the use of grammar and syntax could be improved. These instances often make it difficult to understand the intended meaning of the sentences, which could potentially mislead the readers.
2. To enhance the readability and understanding of the research methodology, I suggest the authors include a flow diagram illustrating the steps and the sequence of their research process. This will provide readers with a visual overview of the methodology and can aid in understanding complex processes.
3. The methods section currently lacks sufficient detail for others in the field to reproduce the experiments. Specific parameters, conditions, statistical tests, and software versions need to be provided to ensure transparency and reproducibility. Providing more detail here would greatly enhance the utility and integrity of the paper.
4. I suggest that the authors enhance the discussion section by comparing and contrasting their findings with those of other studies in the field. This would provide a broader context for their results and highlight the unique contributions of their research.
5. The authors utilize immune infiltration in their analysis, yet they do not provide a clear source for this data. I recommend they clearly state where this data was sourced from, including the database name, version, and any relevant accession numbers.
6. The authors used the GDSC algorithm in their analysis. While this is a powerful tool, its limitations should also be discussed. I recommend the authors include a section discussing the strengths and limitations of the GDSC algorithm in their study context.

Experimental design

Methods should be described with sufficient information to be reproducible by another investigator.

Validity of the findings

The conclusions should be appropriately stated, should be connected to the original question investigated, and should be limited to those supported by the results. In particular, claims of a causative relationship should be supported by a well-controlled experimental intervention. Correlation is not causation.

---

## Round 0.2 · accepted · Accept

Since authors have fullly made revisions according to comments. I think this paper can be accepted for publication.

Reviewer 1 ·

Basic reporting

Authors addressed my previous concerns and I would endorse its publication.

Experimental design

The study is straightforward and well-designed.

Validity of the findings

The data is solid.

Reviewer 2 ·

Basic reporting

No problems any more.

Experimental design

No problems any more.

Validity of the findings

No problems any more.

Additional comments

No problems any more.

Reviewer 3 ·

Basic reporting

The authors have solved all the problem I have proposed. The manuscript is now avaliable for the acceptance.

Experimental design

Original primary research within Aims and Scope of the journal.

Validity of the findings

Impact and novelty not assessed. Meaningful replication encouraged where rationale & benefit to literature is clearly stated.